# BSQ: Exploring Bit-Level Sparsity for Mixed-Precision Neural Network Quantization

**Huanrui Yang, Lin Duan, Yiran Chen & Hai Li**
Department of Electrical and Computer Engineering
Duke University
Durham, NC 27708, USA
`{huanrui.yang, lin.duan, yiran.chen, hai.li}@duke.edu`

## Abstract

Mixed-precision quantization can potentially achieve the optimal tradeoff between performance and compression rate of deep neural networks, and thus, have been widely investigated. However, it lacks a systematic method to determine the exact quantization scheme. Previous methods either examine only a small manually-designed search space or utilize a cumbersome neural architecture search to explore the vast search space. These approaches cannot lead to an optimal quantization scheme efficiently. This work proposes bit-level sparsity quantization (BSQ) to tackle the mixed-precision quantization from a new angle of inducing bit-level sparsity. We consider each bit of quantized weights as an independent trainable variable and introduce a differentiable bit-sparsity regularizer. BSQ can induce all-zero bits across a group of weight elements and realize the dynamic precision reduction, leading to a mixed-precision quantization scheme of the original model. Our method enables the exploration of the full mixed-precision space with a single gradient-based optimization process, with only one hyperparameter to tradeoff the performance and compression. BSQ achieves both higher accuracy and higher bit reduction on various model architectures on the CIFAR-10 and ImageNet datasets comparing to previous methods.

## 1 Introduction

Numerous deep neural network (DNN) models have been designed to tackle real-world problems and achieved beyond-human performance. DNN models commonly demand extremely high computation cost and large memory consumption, making the deployment and real-time processing on embedded and edge devices difficult (Han et al., 2015b; Wen et al., 2016). To address this challenge, model compression techniques, such as pruning (Han et al., 2015b; Wen et al., 2016; Yang et al., 2020), factorization (Jaderberg et al., 2014; Zhang et al., 2015) and fixed-point quantization (Zhou et al., 2016; Wu et al., 2019; Dong et al., 2019), have been extensively studied. Among them, fixed-point quantization works directly on the data representation by converting weight parameters originally in the 32-bit floating-point form to low-precision values in a fixed-point format. For a DNN model, its quantized version requires much less memory for weight storage. Moreover, it can better utilize fixed-point processing units in mobile and edge devices to run much faster and more efficiently.

Typically, model compression techniques aim to reduce a DNN model size while maintaining its performance. The two optimization objectives in this tradeoff, however, have a contrary nature: the performance can be formulated as *a differentiable loss function $\mathcal{L}(W)$* w.r.t. the model's weights $W$; yet the model size, typically measured by the number of non-zero parameters or operations, is *a discrete function* determined mainly by the model architecture. To co-optimize the performance and model size, some previous pruning and factorization methods relax the representation of model size as a differentiable regularization term $R(W)$. For example, group Lasso (Wen et al., 2016) and DeepHoyer (Yang et al., 2020) induce weight sparsity for pruning, and the attractive force regularizer (Wen et al., 2017) and nuclear norm (Xu et al., 2018) are utilized to induce low rank. The combined objective $\mathcal{L}(W) + \alpha R(W)$ can be directly minimized with a gradient-based optimizer for optimizing the performance and model size simultaneously. Here, the hyperparameter $\alpha$ controls the strength of the regularization and governs the performance-size tradeoff of the compressed model.

Unlike for pruning and factorization, there lacks a well-defined differentiable regularization term that can effectively induce quantization schemes. Early works in quantization mitigate the tradeoff exploration complexity by applying the same precision to the entire model. This line of research focuses on improving the accuracy of ultra low-precision DNN models, e.g., quantizing all the weights to 3 or less bits (Zhou et al., 2016; Zhang et al., 2018), even to 1-bit (Rastegari et al., 2016). These models commonly incur significant accuracy loss, even after integrating emerging training techniques like straight-through estimator (Bengio et al., 2013; Zhou et al., 2016), dynamic range scaling (Polino et al., 2018) and non-linear trainable quantizers (Zhang et al., 2018). As different layers of a DNN model present different sensitivities with performance, a mixed-precision quantization scheme would be ideal for the performance-size tradeoff (Dong et al., 2019). There have also been accelerator designs to support the efficient inference of mixed-precision DNN models (Sharma et al., 2018). However, to achieve the optimal layer-wise precision configuration, it needs to exhaustively explore the aforementioned discrete search space, the size of which grows exponentially with the number of layers. Moreover, the dynamic change of each layer's precision cannot be formulated into a differentiable objective, which hinders the efficiency of the design space exploration. Prior studies (Wu et al., 2019; Wang et al., 2019) utilize neural architecture search (NAS), which suffers from extremely high searching cost due to the large space of mixed-precision quantization scheme. Recently, Dong et al. (2019) propose to rank each layer based on the corresponding Hessian information and then determine the relative precision order of layers based on their ranking. The method, however, still requires to manually select the precision level for each layer.

Here, we propose to revisit the fixed-point quantization process from a new angle of *bit-level sparsity*: decreasing the precision of a fixed-point number can be taken as forcing one or a few bits, most likely the least significant bit (LSB), to be zero; and reducing the precision of a layer is equivalent to zeroing out a specific bit of all the weight parameters of the layer. In other words, the precision reduction can be viewed as increasing the layer-wise bit-level sparsity. By considering the bits of fixed-point DNN parameters as continuous trainable variables during DNN training, we can utilize a sparsity-inducing regularizer to explore the bit-level sparsity with gradient-based optimization, dynamically reduce the layer precision and lead to a series of mixed-precision quantization schemes. More specific, we propose Bit-level Sparsity Quantization (*BSQ*) method with the following contributions:

- We propose *a gradient based training algorithm for bit-level quantized DNN models*. The algorithm considers each bit of quantized weights as an independent trainable variable and enables the gradient-based optimization with straight-through estimator (STE).
- We propose a *bit-level group Lasso regularizer* to dynamically reduce the weight precision of every layer and therefore induce mixed-precision quantization schemes.
- BSQ uses only one hyperparameter, the strength of the regularizer, to trade-off the model performance and size, making the exploration more efficient.

This work exclusively focuses on layer-wise mixed-precision quantization, which is the granularity considered in most previous works. However, the flexibility of BSQ enables it to explore mixed-precision quantization of any granularity with the same cost regardless of the search space size.

## 2 RELATED WORKS ON DNN QUANTIZATION

Quantization techniques convert floating-point weight parameters to low-precision fixed-point representations. Directly quantizing a pre-trained model inevitably introduces significant accuracy loss. So many of early research focus on how to finetune quantized models in low-precision configurations. As the quantized weights adopt discrete values, conventional gradient-based methods that are designed for continuous space cannot be directly used for training quantized models. To mitigate this problem, algorithms like DoReFa-Net utilize a straight-through estimator (STE) to approximate the quantized model training with trainable floating-point parameters (Zhou et al., 2016). As shown in Equation (1), a floating-point weight element $w$ is kept throughout the entire training process. Along the forward pass, the STE will quantize $w$ to $n$-bit fixed-point representation $w_q$, which will be used to compute the model output and loss $\mathcal{L}$. During the backward pass, the STE will directly pass the gradient w.r.t. $w_q$ onto $w$, which enables $w$ to be updated with the standard gradient-based optimizer.

$$\textbf{Forward:} \ w_q = \frac{1}{2^n - 1} Round[(2^n - 1)w]; \ \textbf{Backward:} \ \frac{\partial \mathcal{L}}{\partial w} = \frac{\partial \mathcal{L}}{\partial w_q}. \quad (1)$$

Early studies revealed that weights of different layers have different dynamic ranges. It is important to keep the dynamic range of each layer for maintaining the model performance, especially for quantized models. He et al. (2016b) and Polino et al. (2018) propose to explicitly keep track of the dynamic range of each layer by scaling all the weight elements in a layer to the range of [0,1] at every training step, before applying the quantization STE. Other techniques, such as learnable nonlinear quantifier function (Zhang et al., 2018) and incremental quantization (Zhou et al., 2017), are also useful in improving the performance of quantized models. However, it is still very difficult to quantize the entire DNN model to a unified ultra-low precision without incurring significant accuracy loss.

Recent research shows that different layers in a DNN model contribute to the overall performance in varying extents. Therefore mixed-precision quantization scheme that assigns different precision to layers (Wu et al., 2019; Dong et al., 2019) presents a better accuracy-compression tradeoff. The challenge lies in how to determine the quantization scheme, i.e., the precision of each layer, as it needs to explore a large and discrete search space. Some works design quantization criteria based on concepts like "noise gain" (Sakr & Shanbhag, 2018; 2019) to constraint the relationship between each layer's precision and thus largely reduce the search space, yet those criteria are often heuristic, preventing these methods to reach ultra-low precision and find the optimal tradeoff point between model size and accuracy. Other works utilize neural architecture search (NAS). For example, Wang et al. (2019) consider the precision assignment of each layer as an action and seek for the optimal design policy via reinforcement learning. Wu et al. (2019) combine all possible design choices into a "stochastic super net" and approximate the optimal scheme via sampling. However, the cost of NAS methods scales up quickly as the quantization search space grows exponentially with the number of layers. Common practices of constraining the search cost include limiting the precision choices or designing the quantization scheme in a coarse granularity. A recent line of research work attempts to rank layers based on their importance measured by the sensitivity or Hessian information. Higher precision will then be assigned to more important layers (Dong et al., 2019). The exact precision of each layer, however, needs to be manually selected. So these methods cannot adequately explore the whole search space for the optimal quantization scheme.

## 3 THE BSQ METHOD

BSQ aims to obtain an optimal mixed-precision quantization scheme through a single-pass training process of a quantized model. In this section, we first introduce how to convert a DNN model to the bit representation and propose a gradient-based algorithm for training the resulted bit-level model. A bit-level group Lasso regularizer is then proposed to induce precision reduction. In the end, we elaborate the overall training objective of BSQ and the dynamical precision adjustment procedure.

### 3.1 TRAINING THE BIT REPRESENTATION OF DNN

As illustrated in Figure 1(a), we convert a floating-point weight matrix $W$ of a pretrained network to its bit representation through a pipeline of scaling, quantization and binary conversion. Similar to the practice in (He et al., 2016b; Polino et al., 2018), we retain the dynamic range of $W$ by scaling all the elements to the range of $[0, 1]$ before applying quantization. However, these prior works always scale the largest element to $1$ to fully utilize all the quantized bins at every training step, which makes the dynamic precision reduction impossible. Instead, our method conducts the scaling only

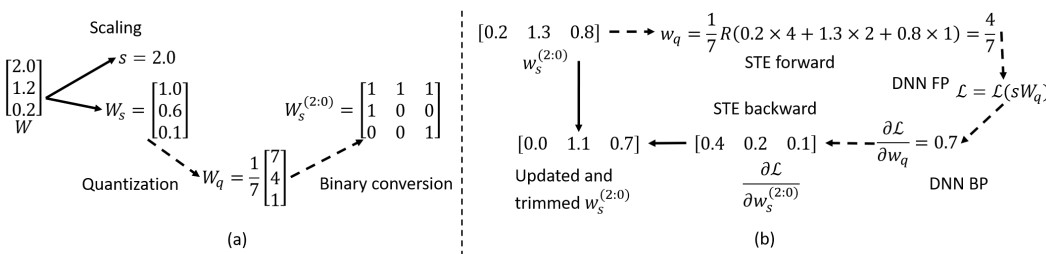

Figure 1: An example of DNN training under the bit representation with precision $n = 3$. (a) Pipeline of converting from the floating-point weight $W$ to the bit representation; (b) Training the bit-level model weight with STE.

once, which is right before the bit representation training. Formally, before converting $W$ to its bit representation, we first extract its dynamic range as $W = s \cdot W_s$, where $s = \max|W|$ is the scaling factor and $W_s$ is the scaled weight matrix. The absolute value of any element $w_s$ in $W_s$ is within the range of $[0, 1]$. Now we apply an $n$-bit uniform quantization to the absolute value of $w_s$ such as $w_q = Round[|w_s| \times (2^n - 1)]/(2^n - 1)$. Then $w_q$ can be exactly represented by a $n$-bit binary number as $w_q = [\sum_{b=0}^{n-1} w_s^{(b)} 2^b]/(2^n - 1)$, where $w_s^{(b)}$ denotes the $b^{th}$ bit in the binary representation. Till this point, $W$ in the floating-point form is replaced with

$$W \equiv sign(W) \odot sW_q \equiv sign(W) \odot \frac{s}{2^n - 1} \sum_{b=0}^{n-1} W_s^{(b)} 2^b, \tag{2}$$

where $\odot$ denotes the element-wise Hadamard product. We consider the bit representation $W_s^{(b)}$ where $b \in [0, n-1]$ and the scaling factor $s$ as independent trainable variables in the training process.

Note that $W_s^{(b)}$ is composed of binary values by definition and $sign(W)$ is a discrete function. Neither of them can be directly trained with gradient descent. To mitigate the binary constraint of $W_s^{(b)}$, we adopt the STE proposed by Bengio et al. (2013) during the training process. As shown in Equation (1), STE enables a quantized model to be trained with continuous floating-point weights. Specifically, the STE for the bit representation training is defined as:

$$\textbf{Forward: } W_q = \frac{1}{2^n - 1} Round \left[ \sum_{b=0}^{n-1} W_s^{(b)} 2^b \right]; \quad \textbf{Backward: } \frac{\partial \mathcal{L}}{\partial W_s^{(b)}} = \frac{2^b}{2^n - 1} \frac{\partial \mathcal{L}}{\partial W_q}. \tag{3}$$

STE relaxes the binary constraint and allows gradient updates for the elements in $W_s^{(b)}$. As illustrated in Figure 1(b), during the forward pass, $s \cdot W_q$ will be used to reconstruct the model weight $W$ and compute the loss, which demonstrates the performance of the current model after quantization. The gradient w.r.t. $W_q$ from the back-propagation will be passed through the rounding function and updated on the continuous values of $W_s^{(b)}$. The proposed bit representation can therefore be trained with any gradient-based optimizer.

The proposed bit representation training only leads to minimal computational and run-time memory overhead comparing to the normal back propagation procedure. From the memory consumption perspective, the bit representation training treats each bit as separated floating-point trainable variables, so a $N$-bit model in bit representation will have $N$ times more parameters and gradients to be stored comparing to that of the baseline training. Though for actual run-time memory consumption, the hidden feature between each layer consumes a significantly larger memory than weights and gradients. As the bit representation does not affect the hidden features, the increase in trainable variables does not lead to significant increase in run-time memory consumption. From the perspective of computation cost, note that the gradient w.r.t. each $W_s^{(b)}$ can be computed as the gradient w.r.t. the corresponding $W_q$ scaled by a power of 2. So under a $N$-bit scheme there will only be $N$ additional scaling for each parameter comparing to the normal training. These additional computations are very cheap comparing to the floating-point operations involved in back propagation. So the proposed bit representation training only leads to minimal computational overhead comparing to a normal back propagation.

We restrict the value of $W_s^{(b)}$ within $[0, 2]$ throughout the training, so that the corresponding $W_q$ has the chance to increase or decrease its precision in the "precision adjustment" step, which will be discussed in Section 3.3. This is enforced by trimming $W_s^{(b)}$ to 0 or 2 if it exceeds the range after a training step.

To enable the dynamic update of $sign(W)$ during training, we separate the positive and negative elements in $W_s$ as $W_s = (W_p - W_n)$ before quantization. Here $W_p = W_s \odot \mathbb{1}(W_s \geq 0)$ contains all the positive elements and $W_n = -W_s \odot \mathbb{1}(W_s < 0)$ includes the absolute value of all the negative weight elements. $W_p$ and $W_n$ will be respectively converted to $W_p^{(b)}$ and $W_n^{(b)}$ by following the process in Equation (2), so that $W_s^{(b)} = W_p^{(b)} - W_n^{(b)}$. Note that the replacement of $W_s^{(b)}$ with $W_p^{(b)} - W_n^{(b)}$ does not introduce any non-differentiable function. Therefore all elements in $W_p^{(b)}$ and $W_n^{(b)}$ can take continuous values between $[0, 2]$ and be trained with the bit representation STE in Equation (3). As such, the original weight matrix $W$ is converted into trainable variables $W_p^{(b)}, W_n^{(b)}$ and $s$ throughout the BSQ training process.

## 3.2 BIT-LEVEL GROUP LASSO

To induce the mixed-precision quantization scheme of a DNN model during training, we propose a bit-level group Lasso ($B_{GL}$) regularizer based on the group Lasso (Hastie et al., 2015) and apply it to $W_p^{(b)}$ and $W_n^{(b)}$ that are converted from a group of weights $W^g$. The regularizer is defined as:

$$B_{GL}(W^g) = \sum_{b=0}^{n-1} \left\| \left[ W_p^{(b)}; W_n^{(b)} \right] \right\|_2, \tag{4}$$

where $W_p^{(b)}$ and $W_n^{(b)}$ are bit representations converted from $W^g$, and $[\cdot; \cdot]$ denotes the concatenation of matrices. $B_{GL}$ could make a certain bit $b$ of all elements in both $W_p^{(b)}$ and $W_n^{(b)}$ zero simultaneously. The bit can thus be safely removed for the precision reduction. Note that the granularity of the quantization scheme induced by $B_{GL}$ is determined by how $W^g$ is grouped. Our experiments organize $W^g$ in a layer-wise fashion. So all elements in a layer have the same precision, which is a common setting in previous mixed-precision quantization work. $W^g$ can also be arranged as any group of weight elements, such as block-wise, filter-wise or even element-wise if needed. Accordingly, the formulation of the regularizer need to be revised to assist the exploration of the mixed-precision quantization at the given granularity. The cost for evaluating and optimizing the regularizer will remain the same under different granularity settings.

## 3.3 OVERALL TRAINING PROCESS

The overall training process starts with converting each layer of a pretrained floating-point model to the bit representation with a relatively high initial precision (e.g., 8-bit fixed-point). BSQ training is then preformed on the achieved bit representation with bit-level group Lasso integrated into the training objective. Re-quantization steps are conducted periodically to identify the bit-level sparsity induced by the regularizer and allow dynamic precision adjustment. As the mixed-precision quantization scheme is finalized, the achieved model is further finetuned for a higher accuracy.

**Objective of BSQ training.** For higher memory efficiency it is desired to find a mixed-precision quantization scheme that minimizes the total number of bits in the model. Thus, in BSQ training we propose to penalize more on the layers with more bits by performing a *memory consumption-aware reweighing* to $B_{GL}$ across layers. Specifically, the overall objective of training a $L$-layer DNN model with BSQ is formulated as:

$$\mathcal{L} = \mathcal{L}_{CE}(W_q^{(1:L)}) + \alpha \sum_{l=1}^{L} \frac{\#Para(W^l) \times \#Bit(W^l)}{\#Para(W^{(1:L)})} B_{GL}(W^l). \tag{5}$$

Here $\mathcal{L}_{CE}(W_q^{(1:L)})$ is the original cross entropy loss evaluated with the quantized weight $W_q$ acquired from the STE in Equation (3), $\alpha$ is a hyperparameter controlling the regularization strength, and $\#Para(W^l)$ and $\#Bit(W^l)$ respectively denote the parameter number and precision of layer $l$. The loss function in Equation (5) enables a layer-wise adjustment in the regularization strength by applying a stronger regularization on a layer with higher memory usage.

**Re-quantization and precision adjustment.** As BSQ trains the bit representation of the model with floating-point variables, we perform *re-quantization* to convert $W_p^{(b)}$ and $W_n^{(b)}$ to exact binary values and identify the all-zero bits that can be removed for precision reduction. The re-quantization step reconstructs the quantized scaled weight $W_q'$ from $W_p^{(b)}$ and $W_n^{(b)}$ as: $W_q' = Round\left[ \sum_{b=0}^{n-1} W_p^{(b)} 2^b - \sum_{b=0}^{n-1} W_n^{(b)} 2^b \right]$. As we allow the values of $W_p^{(b)}$ and $W_n^{(b)}$ to be within $[0, 2]$, the reconstructed $W_q'$ has a maximum absolute value of $2^{n+1}$. In this way, $W_q'$ is converted to a $(n + 1)$-bit binary number, where each bit is denoted by $W_q^{(b)}$. After the re-quantization, we will adjust the precision of each layer. Specifically, we first check $W_q^{(b)}$ from the MSB down to the LSB and remove the bits with all zero elements until the first non-zero bit. The scaling factor $s$ of the layer remains unchanged during this process. A similar check will then be conducted from the LSB up to the MSB. $s$ needs to be doubled when a bit from the LSB side is removed, as all elements in $W_q'$ are shifted right for one bit. Assume that the precision adjustment makes the precision of a layer change from $n$ to $n'$, the scaling factor will be updated as $s' = s \frac{2^{n'}-1}{2^n-1}$. In this way, the bit representations

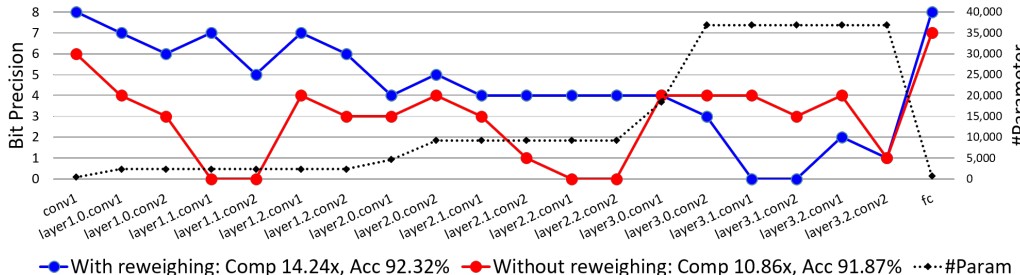

Figure 2: Quantization schemes achieved with or without layer-wise regularization reweighing. The compression rate and the accuracy after finetuning are listed in the legend.

of $W$ before and after the precision adjustment are equivalent, as indicated in Equation (6). The precision of a $n$-bit layer may change between 0 and $(n + 1)$-bit after the precision adjustment.

$$W \equiv \frac{s}{2^n - 1} \sum_{b=0}^{n-1} W_q^{(b)} 2^b \equiv \frac{s'}{2^{n'} - 1} \sum_{b=0}^{n'-1} W_q^{(b)} 2^b. \tag{6}$$

As formulated in Equation (5), the regularization strength assigned to each layer will change with the quantization scheme of the model. The re-quantization and precision adjustment step will be performed periodically during the training process, with an interval of several training epochs. After each precision adjustment, we separate the positive elements and negative elements in $W_q'$ to form the new $W_p^{(b)}$ and $W_n^{(b)}$, respectively. The training can then resume with the newly adjusted $W_p^{(b)}$ and $W_n^{(b)}$ and scaling factor $s'$. It is worth mentioning that $sW_q$ from the forward pass STE remains unchanged before and after the re-quantization and precision adjustment, so the model performance and the gradient from the loss $\mathcal{L}_{CE}$ will not be affected. The interval between re-quantizations needs to be carefully chosen: it shall promptly and properly adjust the regularization strength for stable convergence. The ablation study on re-quantization interval selection is presented in **Appendix B.1**.

**Activation quantization.** Since BSQ modifies only the precision of weights but not affecting the precision of activations, we predetermine the activation precision and fix it throughout the BSQ training process. The activations are quantized in the same way as proposed by Polino et al. (2018). For training stability, we use ReLU-6 activation function for layers with 4-bit or above activations, and use PACT (Choi et al., 2018) for layers with a lower activation precision.

**Post-training finetuning.** At the end of the BSQ training, we perform a final re-quantization and precision adjustment to get the final mixed-quantization scheme. The achieved model can be further finetuned under the obtained precision for improving the overall accuracy. As the quantization scheme is fixed, we adopt the quantization-aware training method proposed by Polino et al. (2018) for finetuning in our experiment.

## 4 ABLATION STUDY

We perform the ablation studies on key design choices of the BSQ algorithm. This section presents the effectiveness of layer-wise memory consumption-aware regularization reweighing and the model size-accuracy tradeoff under different regularization strengths. All experiments are conducted with ResNet-20 models (He et al., 2016a) with 4-bit activation on the CIFAR-10 dataset (Krizhevsky & Hinton, 2009). Detailed experiment setup and hyperparameter choices can be found in **Appendix A**.

### 4.1 EFFECT OF LAYER-WISE REGULARIZATION REWEIGHING

As stated in Equation (5), we propose to apply layer-wise memory consumption-aware reweighing on the $B_{GL}$ regularizer to penalize more on larger layers during the BSQ training. Figure 2 compares the quantization scheme and the model performance achieved when performing the BSQ training with or without such a reweighing term. Here we set the regularization strength $\alpha$ to 5e-3 when training with the reweighing, and to 2e-3 when training without the reweighing to achieve comparable compression rates. All the other hyperparameters are kept the same. As shown in the figure, training without

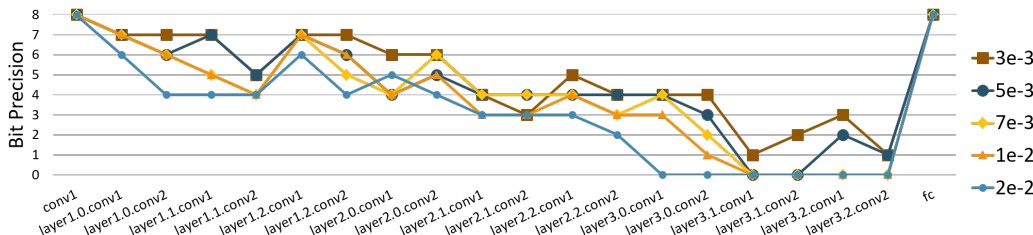

Figure 3: Layer-wise precision comparison of the quantization schemes achieved under different regularization strengths.

Table 1: Accuracy-#Bits tradeoff under different regularization strengths. "FT" stands for finetuning. The last row is achieved by training with quantization schemes achieved by BSQ from scratch.

| Strength $\alpha$ | 3e-3 | 5e-3 | 7e-3 | 1e-2 | 2e-2 |
|---|---|---|---|---|---|
| #Bits per Para / Comp ($\times$) | 3.02 / 10.60 | 2.25 / 14.24 | 1.66 / 19.24 | 1.37 / 23.44 | 0.87 / 36.63 |
| BSQ acc before / after FT (%) | 91.30 / 92.60 | 90.98 / 92.32 | 90.42 / 91.48 | 90.35 / 91.16 | 85.77 / 89.49 |
| Train from scratch acc (%) | 91.72 | 91.45 | 91.12 | 89.57 | 89.14 |

the reweighing term will lead to over-penalization on earlier layers with fewer parameters, while later layers with more parameters are not compressed enough. Therefore the achieved quantized model will have less accuracy even with a smaller compression rate comparing to the model achieved with layer-wise regularization reweighing. As we only show one pair of comparison here, the difference between BSQ training with or without the reweighing term is consistent when varying the regularization strength $\alpha$. Additional results with other $\alpha$ values are shown in **Appendix B.2**.

## 4.2 ACCURACY-#BITS TRADEOFF UNDER DIFFERENT REGULARIZATION STRENGTHS

We fix all the other hyperparameters while varying only the regularization strength $\alpha$ from 3e-3 to 2e-2, to control the tradeoff between the model size and accuracy achieved by BSQ. The quantization schemes achieved by running BSQ with different $\alpha$'s are shown in Figure 3, and the detailed comparison on the compression rate comparing to the 32-bit floating point model (denoted as "Comp") and the validation accuracy (denoted as "Acc") is summarized in Table 1. As shown in Figure 3, the relative ranking of the precision assignment is mostly consistent under different $\alpha$'s, which is consistent with the previous observation that more important layers should be assigned with higher precision. This effect is further illustrated in **Appendix B.3**, where we compare the quantization scheme achieved by BSQ with the layer importance measured in HAWQ (Dong et al., 2019). Furthermore, as $\alpha$ increases, the overall bit reduction increases with the cost of a small performance loss. This tradeoff is also observed on models trained with 2-bit or 3-bit activation as we show their quantization schemes and performances in **Appendix B.4**. Note that some layers achieve 0-bit precision under large regularization strength, indicating all the weights become zero and the layer can be skipped. This is possible as the shortcut connection existing in the ResNet architecture enables the pass of information even if the weights are all zero in some layers. We also note that BSQ not only finds the desired mixed-precision quantization scheme, but also provides a model with higher performance under the same quantization scheme. As shown in Table 1, when training a model with the same quantization scheme as achieved by BSQ using the DoReFa-Net algorithm (Zhou et al., 2016) from scratch, the resulted accuracy is always lower than the BSQ model after finetuning.

## 5 EXPERIMENTAL RESULTS

In this section we compare BSQ with previous state-of-the-art methods. Here, ResNet-20 models are used for the comparison on the CIFAR-10 dataset, and ResNet-50 and Inception-V3 models (Szegedy et al., 2016) are utilized for the experiments on the ImageNet dataset (Russakovsky et al., 2015). The hyperparameters used for BSQ training and finetuning are listed in **Appendix A**. All the compression

Table 2: Quantization results of ResNet-20 models on the CIFAR-10 dataset. BSQ is compared with DoReFa-Net (Zhou et al., 2016), PACT (Choi et al., 2018), LQ-Net (Zhang et al., 2018), DNAS (Wu et al., 2019) and HAWQ (Dong et al., 2019). "MP" denotes mixed-precision quantization.

| | Benchmarks | | | | BSQ | | |
|---|---|---|---|---|---|---|---|
| Act. Prec. | Method | Weight Prec. | Comp (×) | Acc (%) | $\alpha$ | Comp (×) | Acc (%) |
| 32-bit | Baseline | 32 | 1.00 | 92.62 | | | |
| | LQ-Nets | 3 | 10.67 | 92.00 | 5e-3 | 14.24 | 92.77 |
| | DNAS | MP | 11.60 | 92.72 | 7e-3 | 19.24 | 91.87 |
| | LQ-Nets | 2 | 16.00 | 91.80 | | | |
| 4-bit | HAWQ | MP | 13.11 | 92.22 | 5e-3 | 14.24 | 92.32 |
| 3-bit | LQ-Nets | 3 | 10.67 | 91.60 | 2e-3 | 11.04 | 92.16 |
| | PACT | 3 | 10.67 | 91.10 | 5e-3 | 16.37 | 91.72 |
| | DoReFa | 3 | 10.67 | 89.90 | | | |
| 2-bit | LQ-Nets | 2 | 16.00 | 90.20 | | | |
| | PACT | 2 | 16.00 | 89.70 | 5e-3 | 18.85 | 90.19 |
| | DoReFa | 2 | 16.00 | 88.20 | | | |

Table 3: Quantization results of ResNet-50 and Inception-V3 models on the ImageNet dataset. BSQ is compared with DoReFa-Net (Zhou et al., 2016), PACT (Choi et al., 2018), LSQ (Esser et al., 2019), LQ-Net (Zhang et al., 2018), Deep Compression (DC) (Han et al., 2015a), Integer (Jacob et al., 2018), RVQ (Park et al., 2018), HAQ (Wang et al., 2019) and HAWQ (Dong et al., 2019).

| | ResNet-50 | | | | Inception-V3 | | |
|---|---|---|---|---|---|---|---|
| Method | Prec. | Comp (×) | Top1 (%) | Method | Prec. | Comp (×) | Top1 (%) |
| Baseline | 32 | 1.00 | 76.13 | Baseline | 32 | 1.00 | 77.21 |
| DoReFa | 3 | 10.67 | 69.90 | Integer | 8 | 4.00 | 75.40 |
| PACT | 3 | 10.67 | 75.30 | Integer | 7 | 4.57 | 75.00 |
| LQ-Nets | 3 | 10.67 | 74.20 | RVQ | MP | 10.67 | 74.14 |
| DC | 3 | 10.41 | 75.10 | HAWQ | MP | 12.04 | 75.52 |
| HAQ | MP | 10.57 | 75.30 | | | | |
| LSQ | 3 | 10.67 | 75.80 | | | | |
| BSQ 5e-3 | MP | 11.90 | 75.29 | BSQ 1e-2 | MP | 11.38 | 76.60 |
| BSQ 7e-3 | MP | 13.90 | 75.16 | BSQ 2e-2 | MP | 12.89 | 75.90 |

rates reported in Table 2 and Table 3 are compared to the 32-bit floating point model, and all the accuracy reported is the testing accuracy evaluated on models after finetuning.

Table 2 reports the quantization results of ResNet-20 models on the CIFAR-10 dataset. Here we set the activation of the first convolutional layer and the final FC layer to 8 bits while all the other activations to 4, 3 or 2 bits respectively to match the settings of previous methods. The reported 32-bit activation model performance is achieved by finetuning the 4-bit activation model under full precision activation. The exact BSQ quantization schemes of the 4-bit activation models are listed in Figure 3, while those of the 2-bit and 3-bit activation models can be found in **Appendix B.4**. Comparing to previous mixed-precision quantization methods, the model obtained by BSQ with 4-bit activation and $\alpha = 5e$-3 has slightly higher accuracy as the model achieved by HAWQ (Dong et al., 2019), but a higher compression rate (14.24× vs. 13.11×). The same model with 32-bit activation obtains 23% more compression rate with the same accuracy as the model found by DNAS (Wu et al., 2019), with a much less training cost as our method does not involve the costly neural architecture search. The advantage of BSQ is even larger comparing to single-precision quantization methods (Zhou et al., 2016; Choi et al., 2018; Zhang et al., 2018), as BSQ achieves both higher compression rate and higher accuracy comparing to all methods with the same activation precision.

The results of BSQ and previous quantization methods on the ImageNet dataset are summarized in Table 3. The exact BSQ quantization schemes can be found in **Appendix C**. For ResNet models, the activation of the first and the final layer are set to 8 bits while all the other activations are set to 4

bits. For Inception-V3 models the activation of all the layers are set to 6 bits. On ResNet-50 models, BSQ with $\alpha = 5e\text{-}3$ achieves the same top-1 accuracy as PACT (Choi et al., 2018) and 0.5% less top-1 accuracy as the best available method LSQ (Esser et al., 2019) with a higher compression rate ($11.90\times$ vs. $10.67\times$), showing competitive accuracy-compression tradeoff. BSQ can further increase the compression rate of ResNet-50 to $13.90\times$ with $\alpha = 7e\text{-}3$, with only 0.13% top-1 accuracy loss over the "5e-3" model. On Inception-V3 models, BSQ with $\alpha = 2e\text{-}2$ achieves both higher accuracy (75.90% vs. 75.52%) and higher compression rate ($12.89\times$ vs $12.04\times$) comparing to the best previous method HAWQ (Dong et al., 2019). Adopting a smaller $\alpha = 1e\text{-}2$ makes BSQ to achieve 0.7% accuracy improvement trading off ~10% less compression rate comparing to the "2e-2" model.

## 6 Conclusions

In this work, we propose BSQ, which fully explores the accuracy-model size tradeoff of DNN's mixed-precision quantization schemes with a differentiable training algorithm using DNN's bit representation as trainable variables. A bit-level group Lasso regularizer with memory consumption-aware layer-wise reweighing is applied to induce bit-level sparsity, which leads to the dynamic adjustment of each layer's precision and finally a mixed-precision quantization scheme through a single-pass gradient-based training process. This enables BSQ to dynamically produce a series of quantization schemes trading off accuracy and model size and provides models with higher accuracy comparing to training from scratch under the same quantization scheme. We apply BSQ in training ResNet-20 models on the CIFAR-10 dataset and training ResNet-50 and Inception-V3 models on the ImageNet dataset. In all the experiments, BSQ demonstrates the ability to reach both a better accuracy and a higher compression rate comparing to previous quantization methods. Our results prove that BSQ can successfully fill in the gap of inducing a mixed-precision quantization scheme with a differentiable regularizer, so as to effectively explore the tradeoff between accuracy and compression rate for finding DNN models with both higher accuracy and fewer bits.

### Acknowledgments

This work is supported in part by NSF CCF-1910299 and NSF CNS-1822085.

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

## A  HYPERPARAMETER CHOICES IN THE EXPERIMENTS

### A.1  CIFAR-10 EXPERIMENTS

We use ResNet-20 models on the CIFAR-10 dataset (Krizhevsky & Hinton, 2009) to do all of our ablation studies and evaluate the performance of BSQ. The CIFAR-10 dataset can be directly accessed through the dataset API provided in the "torchvision" python package. We do not change the splitting between the training and the test set. Standard preprocessing procedures, including random crop with a padding of 4, random horizontal flip and normalization, are used on the training set to train the model. The validation set is normalized with the same mean and variance as the training set. We implemented ResNet-20 models following the description in (He et al., 2016a), and pretrain the model for 350 epochs. The learning rate is set to 0.1 initially, and decayed by 0.1 at epoch 150, 250 and 325. The weights of all the layers except the batch normalization are then quantized to 8-bit before the BSQ training. The batch normalization layers are kept in the floating-point format throughout the training process. Similar to previous quantization works, we also apply the activation quantization during the training. For 4-bit or above activation precision we replace all the ReLU activation function in the model with the ReLU6 activation function. For lower activation precision we use the trainable PACT activation (Choi et al., 2018) with weight decay 0.0001. These changes will help achieving higher accuracy and better training stability when the activation is quantized as it eliminates extremely large activation values. As BSQ does not consider activation quantization as an objective, we fix the activation precision throughout the BSQ training and the finetuning process.

We start the BSQ training with the 8-bit quantized pretrained model following the process described in Section 3.3. The BSQ training is done for 350 epochs, with the first 250 epochs using learning rate 0.1 and the rest using learning rate 0.01. Unless otherwise specified, the re-quantization and precision adjustment is done every 100 epochs, as well as after the BSQ training is finished to adjust and finalize the quantization scheme. Different regularization strengths $\alpha$ are tried to explore the tradeoff between accuracy and compression rate. The exact $\alpha$ used for each set of experiment is reported alongside the results in the main article. For comparing with previous methods, we further finetune the achieved mixed-precision model with the DoReFa-Net algorithm (Zhou et al., 2016) while fixing the quantization scheme. The finetuning is performed for 300 epochs with an initial learning rate 0.01 and the learning rate decay by 0.1 at epoch 150 and 250. The "train from scratch" accuracy reported in Table 1 is achieved by first quantizing a pretrained floating-point model to the mixed precision quantization scheme achieved by BSQ, then performing DoReFa-Net quantization aware training on the model. The training is done for 350 epochs, with an initial learning rate 0.1 and the learning rate decay by 0.1 at epoch 150, 250 and 325. All the training tasks are optimized with the SGD optimizer (Sutskever et al., 2013) with momentum 0.9 and weight decay 0.0001, and the batch size is set to 128. All the training processes are done on a single TITAN XP GPU.

### A.2  IMAGENET EXPERIMENTS

The ImageNet dataset is used to further compare BSQ with previous methods in Table 3. The ImageNet dataset is a large-scale color-image dataset containing 1.2 million images of 1,000 categories (Russakovsky et al., 2015), which has long been utilized as an important benchmark on image classification problems. In this paper, we use the "ILSVRC2012" version of the dataset, which can be found at `http://www.image-net.org/challenges/LSVRC/2012/nonpub-downloads`. We use all the data in the provided training set to train our model, and use the provided validation set to evaluate our model and report the testing accuracy. We follow the data reading and preprocessing pipeline suggested by the official PyTorch ImageNet example. For training images, we first perform the random sized crop on the training images with the desired input size, then apply random horizontal flipping and finally normalize them before feeding them into the network. We use an input size of $224 \times 224$ for experiments on the ResNet-50, and use an size of $299 \times 299$ for the Inception-V3 experiments. Validation images are resized then center cropped to the desired input size and normalized before used for testing. For both the ResNet-50 and the Inception-V3 model, the model architecture and the pretrained model provided in the "torchvision" package are directly utilized. The first convolutional layer of the ResNet-50 model and the first 5 conventional layers of the Inception-V3 model are quantized to 8-bit, while all the other layers are quantized to 6-bits before the BSQ training. Similar to the CIFAR-10 experiments, the batch normalization layers are kept as floating-point.

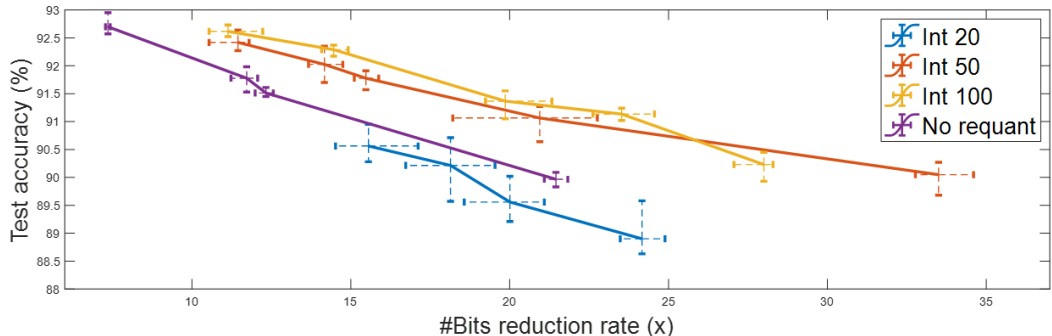

Figure 4: Range of testing accuracy and bit reduction rate achieved from 5 repeated runs with different random seeds. Solid line links the average performance, error bar marks the maximal and minimal performance achieved with each set of hyperparameters.

We start the BSQ training with the quantized pretrained model. For both ResNet-50 and Inception-V3 models, the BSQ training is done for 90 epochs, with the first 30 epochs using learning rate 0.01 and the rest using learning rate 0.001. The re-quantization interval is set to 10 epochs for all the ImageNet experiments. The regularization strength $\alpha$ used is reported alongside the results in Table 3. The model after BSQ training is further finetuned with DoReFa-Net for 90 epochs, with the initial learning rate 0.001 and a learning rate decay by 0.1 after 30 epochs. All the models are optimized with the SGD optimizer with momentum 0.9 and weight decay 0.0001, and the batch size is set as 256 for all the experiments. Two TITAN RTX GPUs are used in parallel for the BSQ training and finetuning of both ResNet-50 and Inception-V3 models.

## B  ADDITIONAL ABLATION STUDY RESULTS

### B.1  CHOICE OF RE-QUANTIZATION INTERVAL

We propose the layer-wise regularization reweighing in Section 3.3 and show its importance in Section 4.1. This reweighing can be more effective if we adjust the precision of each layer regularly throughout the BSQ training routine. The precision adjustment is done through periodic re-quantization. From the one hand, a smaller re-quantization interval would help the precision to be adjusted in-time. From the other hand, it may cause the training unstable due to the frequent change in bit representation and regularizer values. So here we gradually increase the re-quantization interval to find the best choice that can reach high and stable performance. Figure 4 demonstrates the stability and performance under re-quantization intervals 20, 50, 100 and compare them with the performance achieved without re-quantization during the training. Each point in the figure corresponds to the averaged compression rate and accuracy after 5-time repeated BSQ training with a fixed regularization strength $\alpha$ but with different random seeds. The observation in the figure supports our analysis that as re-quantization is important to reach a better accuracy-# bits tradeoff, applying it too frequent will make the training unstable and hinders the overall performance. Comparing to not performing the re-quantization and applying it every 20 epochs, re-quantizing every 50 or 100 epochs yields similarly better tradeoff between accuracy and compression rate. Re-quantization interval 100 leads to a higher accuracy in a wider range of compression rate comparing to the Int 50 model, and the performance is more stable throughout the repeated trails. Therefore in all the other CIFAR-10 experiments we set the re-quantization interval to 100 epochs.

### B.2  ADDITIONAL RESULTS ON REGULARIZATION REWEIGHING

Figure 5 and Figure 6 compares the quantization scheme and the model performance achieved when performing the BSQ training with or without the memory consumption-aware reweighing of the bit-level group Lasso regularizer under additional choices of regularization strength $\alpha$. The $\alpha$ used for each set of experiment are chosen so that comparable compression rates are achieved with or without reweighing. The $\alpha$ used are listed in the caption of the figures. All the other hyperparameters are kept the same. From both figures we can observe a consistent trend that training without the reweighing

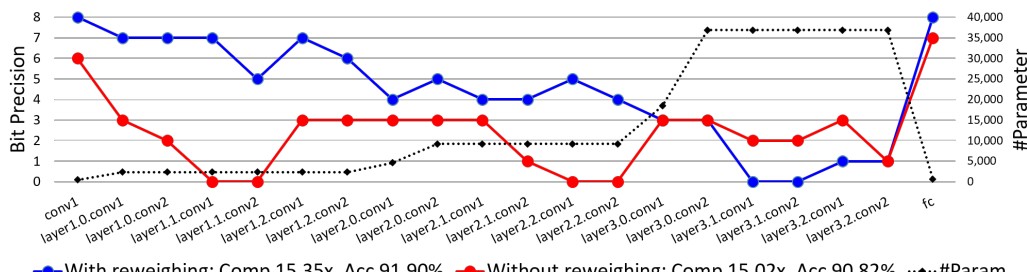

Figure 5: Quantization schemes achieved with or without layer-wise regularization reweighing. The compression rate and the accuracy after finetuning are listed in the legend. $\alpha =$6e-3 with reweighing and $\alpha =$3e-3 without reweighing.

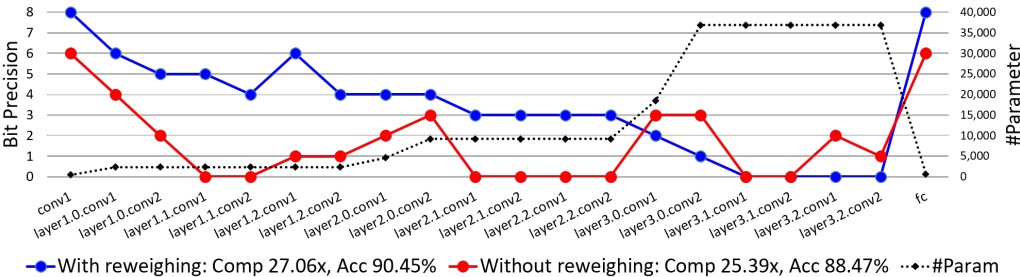

Figure 6: Quantization schemes achieved with or without layer-wise regularization reweighing. The compression rate and the accuracy after finetuning are listed in the legend. $\alpha =$0.015 with reweighing and $\alpha =$5e-3 without reweighing.

term will lead to less precision assigned to earlier layers with fewer parameters, while later layers with more parameters are not compressed enough. Therefore the achieved quantized model will have less accuracy and smaller compression rate comparing to the model achieved with layer-wise regularization reweighing. This observation is consistent with the results shown in Section 4.1. All these results show that the memory consumption-aware reweighing proposed in BSQ training is crucial for generating models with both higher compression rate and higher accuracy.

## B.3 QUANTIZATION SCHEME COMPARISON WITH HAWQ

As discussed in Section 4.2 and shown in Figure 3, for the same model architecture the relative ranking of the precision assignment by BSQ is mostly consistent under different $\alpha$'s. Here we compare quantization schemes achieved by BSQ with the "layer importance ranking" measured in HAWQ (Dong et al., 2019) to further analyze this consistency. HAWQ proposes to rank all the layers in the model with an importance score $S_i = \lambda_i/n_i$, where $\lambda_i$ denotes the top eigenvalue of the Hessian matrix of layer $i$, and $n_i$ represents the number of parameters in layer $i$. A layer with a higher $S_i$ will be assigned with higher precision in the mixed-precision quantization scheme. The quantization schemes achieved by BSQ and HAWQ are compared in Figure 7, where the black dotted

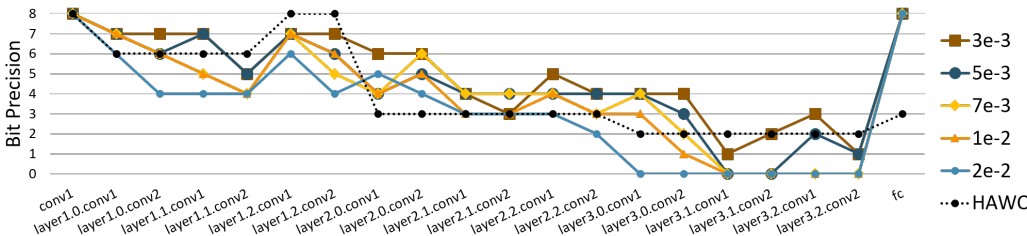

Figure 7: Layer-wise precision comparison between the quantization schemes achieved with BSQ and the scheme achieved with HAWQ (Dong et al., 2019) on the ResNet-20 model.

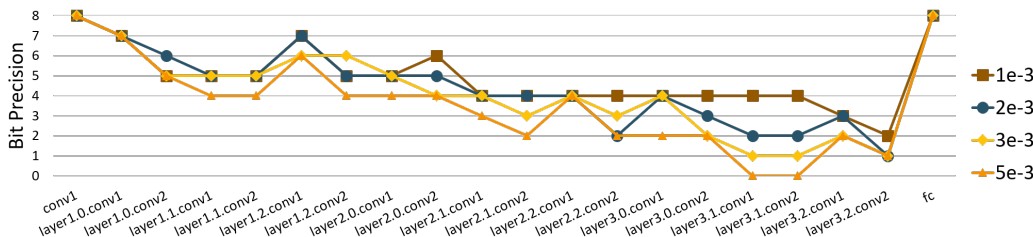

Figure 8: Layer-wise precision comparison of the quantization schemes achieved under different regularization strengths with 2-bit activation.

Table 4: Accuracy-#Bits tradeoff with 2-bit activation. "FT" stands for finetuning.

| Strength $\alpha$ | 1e-3 | 2e-3 | 3e-3 | 5e-3 |
|---|---|---|---|---|
| #Bits per Para / Comp ($\times$) | 3.77 / 8.48 | 2.86 / 11.20 | 2.26 / 14.13 | 1.70 / 18.85 |
| BSQ acc before / after FT (%) | 91.03 / 91.21 | 90.19 / 90.70 | 89.54 / 90.39 | 88.13 / 90.19 |

line shows the HAWQ scheme while the color solid lines demonstrate the schemes achieved by BSQ under different $\alpha$. It can be observed that the relative ranking of BSQ's precision is consistent with the ranking of precision in HAWQ, which to some extent shows that BSQ can dynamically identify the important layers during the training and assign higher precision to them. Note that HAWQ can only come up with the precision ranking of each layer, while the exact precision is designed manually. BSQ on the other hand is able to explicitly assign the precision to each layer during a single training process, and can dynamically tradeoff model size and accuracy by only changing $\alpha$. Thus BSQ can easily find better tradeoff points with both higher accuracy and higher compression rate comparing to HAWQ and other quantization methods, as discussed in Section 5.

### B.4 MODELS ACHIEVED UNDER DIFFERENT ACTIVATION PRECISION

As we discuss the BSQ quantization schemes and model performances under 4-bit activation in Figure 3 and Table 1, here we show the tradeoff between model size and accuracy under different regularization strength $\alpha$ with 2-bit activation in Figure 8 and Table 4, as well as those with 3-bit activation in Figure 9 and Table 5. In both cases we observe that the relative ranking of the precision assignment is mostly consistent under different $\alpha$'s. As $\alpha$ increases, less bits are assigned to each layer, leading to increasing overall bit reduction with the cost of a small performance loss. This tradeoff is consistent with our previous observations on the 4-bit activation models .

## C DETAILED QUANTIZATION SCHEMES FOR IMAGENET EXPERIMENTS

The quantization schemes of the reported ResNet-50 and Inception-V3 models can be found in Table 6 and Table 7 respectively.

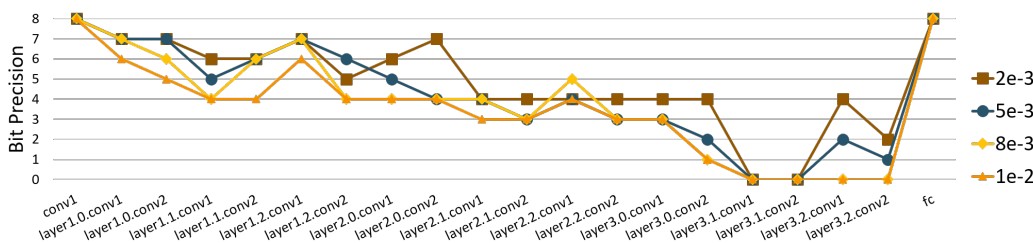

Figure 9: Layer-wise precision comparison of the quantization schemes achieved under different regularization strengths with 3-bit activation.

Table 5: Accuracy-#Bits tradeoff with 3-bit activation. "FT" stands for finetuning.

| Strength $\alpha$ | 2e-3 | 5e-3 | 8e-3 | 1e-2 |
|---|---|---|---|---|
| #Bits per Para / Comp ($\times$) | 2.90 / 11.04 | 1.95 / 16.37 | 1.39 / 23.04 | 1.28 / 25.06 |
| BSQ acc before / after FT (%) | 90.45 / 92.16 | 90.44 / 91.72 | 89.01 / 90.93 | 88.41 / 90.51 |

Table 6: Quantization schemes of ResNet-50 models on ImageNet dataset achieved by BSQ in Table 3. The scheme on the left is achieved with $\alpha = 5e$-3 and the one on the right is achieved with $\alpha = 7e$-3. Except the first row for the leading convolutional layer and the last row for the FC layer, each row in the table reports the precision assigned to the 3 layers in a residual block, with layer 1-3 listed from left to right.

| | BSQ 5e-3 | | | BSQ 7e-3 | | |
|---|---|---|---|---|---|---|
| Conv 1 | 7 | | | 7 | | |
| Block 1-0 | 7 | 6 | 6 | 7 | 6 | 6 |
| Block 1-1 | 6 | 6 | 6 | 6 | 6 | 6 |
| Block 1-2 | 6 | 6 | 6 | 6 | 5 | 6 |
| Block 2-0 | 4 | 3 | 4 | 4 | 3 | 4 |
| Block 2-1 | 4 | 4 | 4 | 4 | 3 | 4 |
| Block 2-2 | 4 | 4 | 4 | 4 | 3 | 4 |
| Block 2-3 | 4 | 3 | 4 | 3 | 3 | 4 |
| Block 3-0 | 4 | 3 | 3 | 4 | 3 | 3 |
| Block 3-1 | 3 | 3 | 4 | 3 | 3 | 3 |
| Block 3-2 | 3 | 3 | 3 | 3 | 3 | 3 |
| Block 3-3 | 3 | 3 | 3 | 3 | 2 | 3 |
| Block 3-4 | 3 | 3 | 3 | 3 | 3 | 3 |
| Block 3-5 | 3 | 3 | 3 | 3 | 3 | 3 |
| Block 4-0 | 3 | 2 | 2 | 3 | 2 | 2 |
| Block 4-1 | 2 | 2 | 3 | 2 | 2 | 2 |
| Block 4-2 | 2 | 3 | 3 | 2 | 2 | 2 |
| FC | 3 | | | 2 | | |

Table 7: Quantization schemes of Inception-V3 model on ImageNet dataset achieved by BSQ in Table 3. The scheme on the left is achieved with $\alpha = 1e$-2 and the one on the right is achieved with $\alpha = 2e$-2. Except for the first 5 convolutional layers and the final FC layer, each row in the table reports the precision assigned to the layers within the inception block. The order from left to right follows the parameter definition order provided in the torchvision package implementation (`https://github.com/pytorch/vision/blob/master/torchvision/models/inception.py`).

| | BSQ 1e-2 | | | | | | | | | | BSQ 2e-2 | | | | | | | | | |
|---|---|---|---|---|---|---|---|---|---|---|---|---|---|---|---|---|---|---|---|---|
| Conv 1a | 8 | | | | | | | | | | 8 | | | | | | | | | |
| Conv 2a | 7 | | | | | | | | | | 7 | | | | | | | | | |
| Conv 2b | 6 | | | | | | | | | | 6 | | | | | | | | | |
| Conv 3b | 8 | | | | | | | | | | 8 | | | | | | | | | |
| Conv 4a | 5 | | | | | | | | | | 4 | | | | | | | | | |
| Mixed 5b | 4 | 4 | 4 | 4 | 4 | 3 | 4 | | | | 4 | 4 | 3 | 4 | 3 | 3 | 4 | | | |
| Mixed 5c | 4 | 4 | 3 | 4 | 3 | 3 | 4 | | | | 4 | 4 | 3 | 4 | 3 | 3 | 4 | | | |
| Mixed 5d | 4 | 4 | 4 | 4 | 4 | 3 | 4 | | | | 4 | 4 | 3 | 4 | 3 | 3 | 4 | | | |
| Mixed 6a | 2 | 4 | 4 | 3 | | | | | | | 2 | 4 | 3 | 3 | | | | | | |
| Mixed 6b | 4 | 3 | 3 | 3 | 3 | 3 | 3 | 3 | 3 | 3 | 3 | 3 | 3 | 3 | 3 | 3 | 3 | 3 | 3 | 3 |
| Mixed 6c | 4 | 4 | 3 | 3 | 4 | 3 | 3 | 3 | 3 | 3 | 3 | 3 | 3 | 3 | 3 | 3 | 3 | 2 | 2 | 3 |
| Mixed 6d | 5 | 3 | 3 | 3 | 4 | 3 | 3 | 3 | 3 | 3 | 5 | 3 | 3 | 3 | 4 | 3 | 3 | 3 | 2 | 3 |
| Mixed 6e | 5 | 4 | 3 | 3 | 3 | 3 | 3 | 3 | 3 | 4 | 4 | 3 | 2 | 2 | 3 | 3 | 3 | 3 | 3 | 3 |
| Mixed 7a | 3 | 3 | 4 | 3 | 3 | 2 | | | | | 3 | 3 | 3 | 3 | 3 | 2 | | | | |
| Mixed 7b | 2 | 3 | 3 | 3 | 2 | 2 | 3 | 2 | 3 | | 2 | 2 | 3 | 3 | 2 | 1 | 2 | 2 | 2 | |
| Mixed 7c | 2 | 2 | 3 | 3 | 3 | 2 | 3 | 3 | 2 | | 2 | 2 | 3 | 3 | 2 | 2 | 3 | 3 | 2 | |
| FC | 3 | | | | | | | | | | 3 | | | | | | | | | |

