# OpenReview forum: "BSQ: Exploring Bit-Level Sparsity for Mixed-Precision Neural Network Quantization"
_ICLR.cc/2021/Conference — ICLR 2021 Poster_

### Official Review · AnonReviewer3 · 2020-10-18
**Learnable Quantization Bits**

**Rating:** 6
**Confidence:** 4

**Review:**

This paper basically proposed to learn the quantization bits (precision) in each layer. Specially, weights are constructed with binary representation as $W_s = \[W_s^1,...,W_s^b\]$. During training, $W_s^i$ is relaxed to $ \in \[0, 2\]$. And a group sparsity is imposed to all $W_s^i$ for all weights in a layer, leading to certain $W_s^i \to 0$, thus cancelling the bit allocation in $i$-th. Experimental results is promising.

Pros:
1. It is interesting to see that weights are represented in binary format, while each bit is trained in a full-precision scheme.

Cons:
1. Training process is intricated: one has to tune the penalty in group sparsity. Also, training is separated in several steps: training and post-training finetuning.

Questions:
1. After determining the quantization bit in ("fake") quantization training (although $W_q$ is quantized but $W_s^i$ is not exactly binary, which is the exactly weight we want) using Eq.5. Author mention in "Re-quantization and precision adjustment" that $W_q^{'}$ is converted to binary value. But how to deal with the precision loss here? i.e. from $W_s^i \in \[0,2\]$ to $\{0, 1\}$
2. Author mentioned that DoReFa-Net is adopted to finetune the trained model. Since DoReFa-Net use tanh to constrain value to $[0,1]$. it seems there is no connection to the proposed quantization scheme (Eq.6). How to exactly finetune ?
3. Why is necessary for $W_s$ to be separated into postive and negative part ($W_p$, $W_n$) in processing ?
4. Since $W_s^i$ is float and trainable, is it necessary to incorporate a trainable $s$ ?

---

> ### Author Response · Authors · 2020-11-12
> **Reply to Reviewer 3**
>
> Thank you for your constructive feedback on our paper! To answer your questions:
>
> 1.	This question is similar to question 2 of reviewer 4. As introduced at the top of page 6, effective weight $W=s W_q$ used in the forward pass STE remains unchanged before and after the re-quantization and precision adjustment. This is ensured by the use of STE in Equation (3) where the forward pass is always computed with the quantized weight, and the change in scaling factor $s$ after precision adjustment, as illustrated in Equation (6). Therefore, there will not be any change to the model performance and the final cross-entropy loss before and after the re-quantization and precision adjustment. Also, note that we allow the n-bit weight to be re-quantized to (n+1)-bit after the re-quantization, so there won’t be any precision loss when converting from $W^{(i)}_{s} \in [0,2]$ to 0,1
>
> 2.	Thanks for pointing this out. It seems like we made a mistake here confusing the STE used in (Polino et al., 2018) with the one used in DoReFa. In this work we use the STE with a linear scaling function as proposed in (Polino et al., 2018) to quantize both weight and activation during the finetuning process. Specifically, weight and activation are first linearly scaled to the range of [0,1] before going through uniform quantization in the STE. We will correct this in the revised version.
>
> 3.	The reason why we separate Wp and Wn is to assist to observe bit-level sparsity when both positive and negative elements are presented in a weight tensor, such that we can ignore the sign of the weight element and only focusing on the bit representation of the absolute values. This will simplify the implementation of re-quantization and precision adjustment. Actually, in the training process with floating-point variables, dividing Wp and Wn may not be required. As the gradient passed from the STE in Equation (3) onto the pair of Wp and Wn are always symmetric, training with separated variables is equivalent to training with Ws = Wp-Wn directly.
>
> 4.	As discussed in Section 3.1, although Ws is float and trainable, it will always be in the range of [0,2] for all the layers and will be quantized uniformly with the same step size, which may not be adequate to capture the difference of dynamic range across different layers, hurting the performance of some layers after quantization. In the meantime, using a scaling factor that always scale the largest element to 1 as done in (Polino et al., 2018) will make the dynamic precision reduction impossible. This is why we keep the scaling factor as a separate trainable variable, allow it to fit into the dynamic range requirement of each layer while not preventing the precision reduction of the bit representation.
>
> Reference: A. Polino, R. Pascanu, and D. Alistarh. Model compression via distillation and quantization. arXiv preprint arXiv:1802.05668, 2018.

---

> > ### Comment · AnonReviewer3 · 2020-11-16
> > **Confusion on Question 1**
> >
> > In my understanding, from Eq.2, $W_s^b$ must be $\in { 0, 1} $ to ensure $W_q$ to be integer (aka quantized value). Since $W_s^b$ is trained in float-point, how to convert $[0, 2] \to {0, 1}$? Eq.3 illustrates the quantization process, which seems not require $W_s^b$ to be binary, for later Round operation will lead to a quantized $W_q$. If so, what's the point of setting $W_s^b$.
> >
> > I checked again and I found the symbols used is somehow confusing: actually pipeline goes as: $W \to W_s \to W_q \to W_s^b$. Author should clarify the used symbols. Aagin, if $W_s^b$ not necessarily to be ${0, 1}$, what's the point pf setting $W_s^b$? That's my concerns on the first quenstion.
> >
> > Let me know if I have anything misunderstood.

---

> > > ### Author Response · Authors · 2020-11-16
> > > **Clarification of the notation and the training process**
> > >
> > > Thanks for your questions!
> > >
> > > Note that in BSQ, we consider bit representation $W_s^b$ as trainable variables throughout the training process, so that we can apply the bit-level group lasso regularization on each bit to induce precision reduction.
> > >
> > > To enable gradient-based training, we adopt the idea of straight-through estimator (STE) following previous quantization-aware training work like (Polino et al., 2018), where we keep a set of floating-point "shadow weights" for gradient accumulation, while using quantized weight to perform forward and backward pass computation to make the training process quantization aware.
> > >
> > > Specifically, as illustrated in Equation (3), in the forward pass STE we first reconstruct the floating point shadow weight $W_s = \sum_{b=0}^{n-1} W_s^b 2^b$ from  $W_s^b \in [0,2]$, then the forward pass is computed with $W_q$, the quantized version of $W_s$, as $W_q = \frac{1}{2^n-1} Round[W_s]$. It is the rounding function in the forward pass STE that make sure $W_q$ is a quantized value; while $W_s^b$ (and correspondingly $W_s$) are always kept as floating-point throughout the training process, except for the re-quantization step where we have to assign the value of each $W_s^b \in 0,1$ according to the quantized $W_q$ so as to identify all-zero bits to be removed.
> > >
> > > As for Equation (2), it is only intended to show the relationship between the proposed trainable variables $s$, $W_s^b$ and the original weight matrix $W$. The conversion pipeline is only used to initialize the bit representation from a pretrained floating-point model, then there won't be any conversion from $W$ as we directly use $s$ and $W_s^b$ as trainable variables.

---

> > > > ### Comment · AnonReviewer3 · 2020-11-17
> > > > **Response to Clarification**
> > > >
> > > > Thanks for your clarification.
> > > >
> > > > What I am thinking in beginning reading is that, $W_s^b$ is constrainted to $0, 1$ to realize quantization. So that normal quantization (using round function, as indicated in Eq.3) is circumvented. However, in your clarification, it seems that normal quantization is still in use, which is less interesting as I expect.
> > > >
> > > > The point of setting $W_s^b$(or $W_n^b$, $W_p^b$) is to find out bit to be cancelled for precision reduction. But in Re-quantization, $W_s^b$ is binarized. If the reason to to find out the zero bit, why is $W_s^b$ binarized? It could be sparsified to fullfil your goal.

---

> > > > > ### Author Response · Authors · 2020-11-17
> > > > > **Thanks for the discussion**
> > > > >
> > > > > Thank you very much for these valuable suggestions!
> > > > >
> > > > > We agree with you that constraining each bit individually would be an interesting extension from our current method, though it may need stochastic relaxation techniques like hard concrete distribution (Louizos et al., 2017) to enable the optimization process. We would like to further explore this track as future work. As for the current method, we believe that the proposed idea of inducing mixed-precision quantization scheme via bit representation training and bit-level sparsification is novel and provides interesting contribution to the field.
> > > > >
> > > > >  As for the need of re-quantization, since $W_s^b$ is between [0,2], it may contribute to both higher and lower bit when being quantized to binary. (e.g. [0.5, 0, 1] will become [0, 1, 1] after re-quantization) Therefore when desining the training process we think it is important to first re-quantize the model to binary before removing any bits, as it will reflect the true bit-level sparsity in a quantized model. After thinking about your suggestion, we believe it is possible to only perform precision adjustment but not re-quantization duing the training process. This will still effectively lead to precision reduction, and could potentially make the convergence smoother as the value of trainable variables will not be changed. We will explore this as an extension from BSQ in future works.
> > > > >
> > > > > Reference: Louizos, Christos, Max Welling, and Diederik P. Kingma. "Learning Sparse Neural Networks through $ L_0 $ Regularization." arXiv preprint arXiv:1712.01312 (2017).

---

> > > > > > ### Comment · ~Chaojian_Li1 · 2021-02-27
> > > > > > **More discussion on the re-quantization**
> > > > > >
> > > > > > Dear Authors of BSQ,
> > > > > > Could I add some more questions following the discussion about the re-quantization here?
> > > > > > Based on the example, "[0.5, 0, 1] will become [0, 1, 1] after re-quantization" provided by the authors in the last reply. I am thinking:
> > > > > > 1) Will the re-quantization make the bit sparsity introduced by LASSO regularization ruined?
> > > > > > 2) Will the re-quantization break the "layer-wise bit precision" rule that all weights in the same layer have the same bit precision?
> > > > > >
> > > > > > For 1., if we look at the example of a 3-bits value before re-quantization, [1.2, 1.1, 0]. It will become [1, 1, 1] after re-quantization (1.2*4+1.1*2 = 7.0 -> 7 -> [1, 1, 1]). So in this case, there will no sparsity after re-quantization, so it seems that the LASSO regularization has no or unclear effects on the weights after re-quantization, but such re-quantized weights will be used for final deployment.
> > > > > >
> > > > > > For 2., if we look at the example of two 3-bits value in the same layer before the re-quantization, [0.2, 1.3, 0] and [0.3, 1.4, 0]. They will become [0, 1, 1] and [1, 0, 0] after re-quantization (0.2*4+1.3*2 = 3.4 -> 3 -> [0, 1, 1] ; 0.3*4+1.4*2 = 4.0 -> 4 -> [1, 0, 0]) . And based on the precision-adjustment described in the paper, they will finally become [1, 1] (2-bits) and [1] (1-bit), respectively, which doesn't match the claim of "layer-wise bit precision" in the paper and may become "element-wise bit precision".
> > > > > >
> > > > > > Thank you!

---

> > > > > > > ### Comment · ~Huanrui_Yang1 · 2021-02-27
> > > > > > > **More discussion on the re-quantization**
> > > > > > >
> > > > > > > Hi Chaojian! Thanks for your interest in our work! For your first question, it is true that requantization will change the sparsity pattern in the bit representation. However, with the help of group lasso on bit representation, a certain bit of all the parameters in the layer will  get close to zero simultaneously during the training. Then the requantization serves as a "pruning mechanism" to make the bit exactly zero, thus ready for removal. Though this process may reduce the sparsity of lower bits, it will increase the sparsity of higher bits, helping us removing more bits from the MSB side in the precision adjustment. So in summary, lasso helps making all elements small, and requantization helps enforce strict sparsity, especially from the MSB side.
> > > > > > >
> > > > > > > As for your second question, we perform precision adjustment after the requantization, and will only remove a bit if in all weight elements that bit is zero (that's why we use group lasso instead of regular lasso to induce such simultaneous sparsity). So we can ensure the precision within each layer is consistent.
> > > > > > >
> > > > > > > Speaking of this, we do observe that as percision adjustment helps us finding a better quantization scheme, requantization introduces some instability into the training process. From our previous discussion with the reviewer, we believe it is possible to only perform precision adjustment but not re-quantization duing the training process, such as replacing requantization with a simple pruning step with small threshold. This could be an interesting future direction of this work.

---

### Official Review · AnonReviewer1 · 2020-10-27
**A technique to serialize multi-bit computations and focus on generating high accuracy binary networks using sparsity**

**Rating:** 6
**Confidence:** 4

**Review:**

Quantization of weights in DNNs is a very effective way to reduce the computational and storage costs which can enable deployment of deep learning at the edge. However, determining suitable layer-wise bit-widths while training is a difficult task due to the discrete nature of the optimization problem. This paper proposes to utilize bit-level sparsity as a proxy for bit-width and employ regularization techniques to formulate the problem so that precision can be reduced while training the model.

The method proposed by the authors is sound. It leverages insights that have been employed in a neighboring area (pruning via regularization) and re-purposes those to the problem of quantization. The empirical evaluation is robust as well.

One issue I have with the proposed method is that the parameter space is expanded by a large amount. Since for every scalar weight, we end up with a collection of binary weights. Doesn't this make training more difficult? It would be nice to discuss this issue. And more importantly how does the extra effort compare to other approaches (such as Dorefanet and others).

Regarding the proposed regularization technique. Lasso (least absolute shrinkage and selection operator) is, as far as I am aware, an optimizer that regularizes the L_1 norm of the parameters. Why is the regularizer in eq. (4) using the L_2 norm? Maybe I am missing something and/or there is an inconsistency is the notation/wording.

The authors do a good job comparing with related works. However, one of the main early claims is that all works trying to find per-layer precision do so manually. This is not true, there have been some works that have done exactly that. One example is [1] which analytically determines precisions at all layers using a noise gain concept. It would be nice to contrast with such works as well.

Minor issue:
'comp x' is used in the results (tables) without being defined. It appears to indicate 'compression ratio'. This has to be explicitly stated at least once (maybe in the captions).

references:
[1] Sakr, Charbel, and Naresh R. Shanbhag. "Per-tensor fixed-point quantization of the back-propagation algorithm." ICLR 2019.

---

> ### Author Response · Authors · 2020-11-12
> **Reply to Reviewer 1**
>
> We would like to thank your feedback on our paper and are glad that you find our paper technically sound.
>
> **For the issue on parameter space:** As illustrated in Equation (3) and discussed below the equation, the gradient of the cross-entropy loss w.r.t. each bit of the scaled weight $W_s$ is not independent. In fact, the gradient w.r.t. each $W^{(b)}_s$ has a linear relationship with the gradient w.r.t. the corresponding $W_q$. Thus, the proposed bit representation training only leads to minimal computational overhead comparing to the normal backpropagation procedure.
>
> **For regularization:** As introduced in Section 3.2, here we use a form of group Lasso to induce a structural sparsity such that a certain bit of all elements in the weight tensor can become zero simultaneously. This is enabled by applying an L1 regularization (i.e. sum) across the L2 norm of the group of variables corresponding to each bit. Group Lasso is a well-known regularizer for inducing structural sparsity, and has been applied for DNN compression as in (Wen et al., 2016). We have added a citation to the group Lasso in the revision to make it clearer
>
> Reference: Wei Wen, Chunpeng Wu, Yandan Wang, Yiran Chen, and Hai Li. Learning structured sparsity in deep neural networks. In Advances in neural information processing systems, pp. 2074–2082, 2016.
>
> **For the related work:** Thanks for bringing up this interesting work. We have added it to the related work discussion in the revision. [1] shares a similar method as the related work mentioned by reviewer 2. The method is based on manually designed quantization criteria, which may not lead to the best tradeoff between model size and accuracy, especially after quantization-aware training. Consequently, the weight precision achieved in [1] is much higher than that of BSQ. On the other hand, BSQ induces the mixed-precision quantization scheme during the training process without any heuristic constraints. Therefore, BSQ can fully explore the design space of mixed-precision quantization and find better tradeoff points between model size and accuracy.
>
> Thanks for pointing out the issue of “Comp”. We add the description of it in Section 4.2 where it is first mentioned in the article.

---

> > ### Comment · AnonReviewer1 · 2020-11-23
> > **Unconvincing argument - please address, it's an easy fix**
> >
> > To the authors:
> >
> > This answer unfortunately is not convincing. Eq. (3) clearly indicates one gradient to be computed for every "bit" representation of the weights meaning that compute and storage get multiplied at training time by a factor equal to the number of bits considered. Yes, the expressions among corresponding "bit" weights and gradients are similar, meaning the back-prop compiler should be able to handle them as argued in the text. But that's not the issue, it is clear that the number of parameters to store and number of operations blows up at training time. At any rate, the authors have provided a very qualitative answer to this question. It would be nice to provide a more quantitative one, especially since this calculation is not difficult.
> >
> > To be more explicit:
> > 1) Can you count the number of parameters stored at training time per back-prop iteration and compare to the baseline floating-point training? Please provide a quantitative answer in your manuscript in the form of a number or formula (if trends are more meaningful).
> > 2) Can you count the number of OPs (or multiplications) per back-prop iteration and compare to the baseline floating-point training? Please provide a quantitative answer in your manuscript in the form of a number or formula (if trends are more meaningful).
> >
> > Note that I am still recommending acceptance of this paper but that is with the understanding that this issue will be fixed - I believe it is easy to fix, a simple counting exercise. Please address this issue!

---

> > > ### Author Response · Authors · 2020-11-23
> > > **Thanks for your follow-up question**
> > >
> > > We would like to thank you for this follow up question. To address your concern we provide some analysis in this reply. We will also add these contents to Section 3.1 to make the discussion on complexity more complete.
> > >
> > > 1. For the number of parameters, since we are storing each bit as separated trainable variables and train them as floating-point weight, the amout of trainable parameters scales up linearly with the precision of the model. A $N$-bit model in bit representation will have $N$ times more parameter and gradient storage comparing to baseline training. Though from the prespective of actual runtime memory occupation, the hidden feature between each layer consumes a significantly larger memory than weights and gradients, especially when training with comonly used batchsize like 256. (For example when training ResNet 50 on ImageNet feature consumes >99% runtime memory with batchsize 256.)  So in the common setting where we initialize the bit representation with 8 bits, there would be only ~8% increase in the runtime memory, which is not a significant increase.
> > >
> > > 2. For the number of OPs, this follows our explaination in the paper and in our previous reply. Note that as mentioned in Equation (3), we first gather all the bits into $W_q$ then do the forward pass computation, and we compute the gradient w.r.t. each bit from the gradient w.r.t. $W_q$ in back propagation. So there will only be $N$ additional OP for each parameter comparing to the baseline model under a $N$-bit scheme. Note that the number of OPs in the modern CNN model is typically 100x more than the number of parameters. Moreover, these additional OPs introduced by the bit representation is simply a scaling with a power of 2, which can be computed in a much cheaper way than a typical floating-point multiplication by the computer. Therefore we say bit representation training only leads to minimal computational overhead comparing to the normal backpropagation procedure.
> > >
> > > Reference on memory and OP estimation: <https://github.com/albanie/convnet-burden/blob/master/README.md>

---

> > > > ### Comment · AnonReviewer1 · 2020-11-24
> > > > **Thanks for your answer**
> > > >
> > > > Thank you for your answer. We conclude that there is an overhead that you are able to quantify. Please include a summary of your explanation above to your manuscript. I hope you agree that this discussion is important to your method's description. The reader will accurately understand its trade-offs.

---

> > > > > ### Author Response · Authors · 2020-11-24
> > > > > **The manuscript has been updated**
> > > > >
> > > > > Yes we definitely agree this discussion makes the description of the bit representation training more complete. Thank you for bringing this up! We have added a paragraph in Section 3.1 to reflect this discussion.

---

### Official Review · AnonReviewer4 · 2020-10-28
**Good paper, more analysis required**

**Rating:** 6
**Confidence:** 3

**Review:**

This paper introduces a new method to quantize neural networks in a differentiable manner. Proposed method applies the group lasso on the bit-planes of the weight parameters to let certain LSBs in each layer to be zero-ed out. STE is used to train the binary representation of each bit-plane and the sign of weights during the training. Results demonstrate that the proposed method can achieve higher accuracy and compression ratio compared to previous studies.

I think that the idea of introducing group lasso to prune the entire bit-plane is very interesting and the paper is well written, but some additional analysis will be helpful.

1. I think the result must be compared with more recent papers, such as LSQ (Esser, Steven K., et al., 2020). For example, LSQ demonstrates that it acheives 75.8% top-1 and 92.7% top-5 accuracy with 3/3-bit model (weight/activation) on ResNet-50. However, according to the appendix C, proposed method seems to achieve only 92.16% when the activation is quantized to 3-bit.
2. This is more of a question than suggestion: after the requantization, would the batch-normalization layers function correctly? It seems like the parameters of batch normalization layers are kept the same after the requantization, while the requantization will impact the distribution of activations. Analysis on the difference between the distribution of the activation before the batch normalization layer before and afther the requantization will be helpful to see if the distribution of the activations really do not differ that much, or the batch-norm layer will just adapt to the occasion.

---

> ### Author Response · Authors · 2020-11-12
> **Reply to Reviewer 4**
>
> We would like to thank your feedback on our paper and are glad that you find our paper interesting. For your questions:
>
> 1.	Thanks for bringing up the results in LSQ. The result mentioned in the review is evaluated with 3-bit ResNet-50 model on ImageNet, where LSQ achieves 75.8% top-1 accuracy with a 10.67x compression rate comparing to the full-precision model. As reported in Table 3, BSQ can achieve 75.3% top-1 accuracy with a higher 11.90x compression rate on the ResNet-50 model, so the result is still competitive. We have added this comparison to Table 3 in the paper.
>
> As we consider the scale s as trainable variables in BSQ, it is similar to the training process proposed in LSQ. Although we haven’t made a dedicated analysis on the gradient w.r.t. s as done in LSQ. It would be interesting to further improve BSQ with the update rule proposed in LSQ during the training process in future work.
>
> 2.	As introduced at the top of page 6, the effective weight $W=s W_q$ used in the forward pass STE remains unchanged before and after the re-quantization and precision adjustment. This is ensured by the use of bit representation STE in Equation (3), where the forward pass is always computed with quantized weight, and the adjustment of scaling factor s after precision adjustment as illustrated in Equation (6). Therefore, there will not be any change to the activation and the final cross-entropy loss before and after the re-quantization and precision adjustment, and will not affect the functionality of batch norm layers.

---

### Official Review · AnonReviewer2 · 2020-10-28
**The paper proposes a method to implement each layer with different precision (mixed-precision quantization). The method employed is referred to as bit-level sparsity quantization whereby each bit of the parameter set is treated as a trainable parameter. Overall a well-written paper with a solid reasoning behind the work. The results improve marginally over SOTA methods.**

**Rating:** 7
**Confidence:** 4

**Review:**

The paper proposes a method to implement each layer with different precision (mixed-precision quantization). The method employed is referred to as bit-level sparsity quantization whereby each bit of the parameter set is treated as a trainable parameter. A differential bit sparsity regularizer enables a smooth trade-off between accuracy and complexity/compression.

1) results are a slight improvement over SOTA methods. This is to be expected given the maturity of this topic.
2) typo in Table 2 ("wight precision")

3) It will be good to relate this work to [1] that also studies mixed-precision quantization using a pre-trained floating point network.

[1] Sakr et al., An analytical method to determine minimum per-layer precision of deep neural networks.

Overall a well-written paper with a solid reasoning behind the work. The results improve marginally over SOTA methods.

---

> ### Author Response · Authors · 2020-11-12
> **Reply to Reviewer 2**
>
> Thanks for your positive feedback on our paper! We are glad that you find the paper well-written, and have fixed the typo in the table.
>
> We thank the reviewer for bringing up the related work [1], and have added it to the discussion of the related work. [1] proposes a pre-layer precision assignment framework to quantize pretrained DNN models, the precision assignment is done with manually designed criteria assuming that each layer should contribute equally to the overall noise gains. Although the assumption largely reduces the search space, it may not lead to the optimal tradeoff point between model size and accuracy in practice. Also, directly quantizing a pretrained floating-point model may not lead to the best accuracy as the model weight is not aware of the quantization. This can be seen as [1] requires a much larger average weight precision than BSQ as well as other works to maintain high accuracy.
>
> On the other hand, BSQ induces the mixed-precision quantization scheme during the training process without any heuristic constraints. Therefore, BSQ can fully explore the design space of mixed-precision quantization and find better tradeoff points between model size and accuracy. The training process with bit representation STE also ensures that the model weight is aware of the low-precision quantization, further improving the performance under ultra-low precision. This enables BSQ to achieve a much lower average precision with similar accuracy.

---

### Decision · Program_Chairs · 2021-01-07
**Final Decision**

**Decision:**

Accept (Poster)

**Comment:**

The paper explores a solution for mixed precision quantization. The authors view the weights in their binary format, and suggest to prune the bits in a structured way. Namely, all weights in the same layer should have the same precision, and the bits should be pruned from the least significant to most significant. This point of view allows the authors to exploit techniques used for weight pruning, such as L1 and group lasso regularization.

Although the field of quantization and model compression/acceleration is quite mature by now and has a large body of works, this paper is novel in its approach. Although the improvements provided over SoTA results are not very large, I believe that the novelty of the approach would make this paper a welcome addition to ICLR.

There are a few issues to be dealt with pointed out by the reviewers such as confusing terminology or required clarifications, but these are minor revisions that I trust the authors will be able to add to their paper.